# Effect of Resisted Sprint and Plyometric Training on Lower Limb Functional Performance in Collegiate Male Football Players: A Randomised Control Trial

**DOI:** 10.3390/ijerph18136702

**Published:** 2021-06-22

**Authors:** Shahnaz Hasan, Gokulakannan Kandasamy, Danah Alyahya, Asma Alonazi, Azfar Jamal, Radhakrishnan Unnikrishnan, Hariraja Muthusamy, Amir Iqbal

**Affiliations:** 1Department of Physiotherapy, College of Applied Medical Sciences, Majmaah University, Majmaah 11952, Saudi Arabia; sh.ahmad@mu.edu.sa (S.H.); d.alyahya@mu.edu.sa (D.A.); a.alonazi@mu.edu.sa (A.A.); r.unnikrishnan@mu.edu.sa (R.U.); h.muthusamy@mu.edu.sa (H.M.); 2School of Health and Life Sciences, Teesside University, Middlesbrough TS1 3BA, UK; 3Health and Basic Science Research Centre, Majmaah University, Majmaah 11952, Saudi Arabia; azfarjamal@mu.edu.sa; 4College of Applied Medical Sciences, King Saud University, Riyadh 11443, Saudi Arabia; physioamir@gmail.com

**Keywords:** strength, plyometric training, function performance, resisted sprint training, young football players

## Abstract

The main objectives of this study were to evaluate the short-term effects of resisted sprint and plyometric training on sprint performance together with lower limb physiological and functional performance in collegiate football players. Ninety collegiate football players participated in this three-arm, parallel group randomized controlled trial study. Participants were randomly divided into a control group and two experimental groups: resisted sprint training (RST) (*n* = 30), plyometric training (PT) (*n* = 30), and a control group (*n* = 30). Participants received their respective training program for six weeks on alternate days. The primary outcome measures were a knee extensor strength test (measured by an ISOMOVE dynamometer), a sprint test and a single leg triple hop test. Measurements were taken at baseline and after 6 weeks post-training. Participants, caregivers, and those assigning the outcomes were blinded to the group assignment. A mixed design analysis of variance was used to compare between groups, within-group and the interaction between time and group. A within-group analysis revealed a significant difference (*p* < 0.05) when compared to the baseline with the 6 weeks post-intervention scores for all the outcomes including STN (RST: *d* = 1.63; PT: *d* = 2.38; Control: *d* = 2.26), ST (RST: *d* = 1.21; PT: *d* = 1.36; Control: *d* = 0.38), and SLTHT (RST: *d* = 0.76; PT: *d* = 0.61; Control: *d* = 0.18). A sub-group analysis demonstrated an increase in strength in the plyometric training group (95% CI 14.73 to 15.09, *p* = 0.00), an increase in the single leg triple hop test in the resisted sprint training group (95% CI 516.41 to 538.4, *p* = 0.05), and the sprint test was also improved in both experimental groups (95% CI 8.54 to 8.82, *p* = 0.00). Our findings suggest that, during a short-term training period, RST or PT training are equally capable of enhancing the neuromechanical capacities of collegiate football players. No adverse events were reported by the participants.

## 1. Introduction

Resisted sprint and plyometric training are efficient methods for improving strength and sprint performance and reducing the likelihood of injuries [1]. Significantly, enhanced physical fitness is a prerequisite for motor competence and the acquisition of technical skills to improve functional performance that relies heavily on explosive leg power [1]. Explosive strength is developed through movement-specific training regimes. A sprint event (50 m) consists of acceleration and maximum speed phases. The acceleration phase relies on a powerful extension of the leg joints (gluteus maximus), whereas the maximum speed phase requires the backward and forward rotation of the legs (hamstrings) [2]. The muscular action of the extensors of the hip (gluteus maximus and hamstrings), knee (quadriceps), and ankles (gastrocnemius and soleus) are required during the acceleration phase of sprinting [2]. Some researchers have reported a correlation between maximum absolute isometric strength and maximum speed and have shown that this is significantly related to strength qualities [3,4]. The explosive power improved with stretch-shortening cycles called plyometrics [5]. Plyometric training involves an immediate shortening (concentric contraction) after pre-stretching (eccentric contraction) the active muscle and has traditionally been used for various sports which are dependent on speed and power. Plyometric training can show improvements in leg strength and muscle power [6], acceleration (sprint) [7], running performance [7,8] and can lead to an increase in agility [8]. According to Oxfeldt, training regimes can improve jumping and sprint performance, and can reduce the likelihood of injuries occurring [9]. Previous research on supporting training regimes that aid in the functional performance of football players found that the combination of both plyometric training and resisted sprint training have a positive impact on the maximal strength and sprint performance of football players [10,11]. However, the research has generally focused on a wider range of training methods such as interval training and the Nordic hamstring exercise [11]. The findings of Augustine Gnanaraj indicated that plyometric training and resisted sprint training provided good results on the explosive power of the sprinter but did not indicate which out of the two were better. In other sports such as running, it has been found that plyometric training has a greater impact on sprint performance when compared to resisted sprint training [12].

Similarly, plyometric training has also been found to be beneficial in improving a range of sport-specific skills including sprint performance. In a systematic review, Oxfeldt found surmounting evidence that plyometric training was beneficial in improving sprint performance [11]. Similar to many other studies, this finding found that plyometric training can be beneficial in improving sprint performance, with only a few disagreeing with this statement [13,14]. However, Houghton evaluated plyometric training with a small sample group (*n* = 15); this study could not find significant data to suggest that plyometric training was beneficial in improving sprint performance [15].

In a study surrounding American football, it was found that both plyometric training and resisted sprint training can be used to improve sprint performance [15]. However, there is no indication as to which is more beneficial. To reduce the likelihood of injury, it has been found that plyometric training has been included in injury prevention protocols to improve landing mechanics and joint loading ability [9,16]. In addition, plyometric training has also been beneficial in reducing the likelihood of knee injuries occurring in female athletes at both amateur and elite level [17]. Additionally, sprint training was highly beneficial in reducing the rate of muscular injury in elite footballers when compared with several different training modalities such as plyometrics [18,19,20,21]. However, resisted sprint training did not show conclusive results; it proved to be extremely beneficial in some cases, but in others it did not reduce the injury rate [10].

However, studies evaluating the effects of resisted sprint training and plyometric training on the muscle function and performance of quadriceps are lacking. Thus, this study aimed to evaluate the impact of both plyometric training and resisted sprint training on the muscle strength of knee extensors and the sprint performance of the lower limbs of collegiate football players.

## 2. Materials and Methods

### 2.1. Study Design

A three-arm parallel-group randomized controlled trial study design was used to test the hypothesis, which comprised of a six-week intervention (Figure 1). The study was conducted between 30 November 2020 and 28 March 2021 at the physiotherapy and rehabilitation center and University Football stadium at Majmaah University, Al Majmaah, Riyadh, Saudi Arabia. Collegiate football players were recruited through a local university and sporting clubs and were screened by a senior physiotherapist who had more than 15 years of experience in the assessment and management of sport and musculoskeletal disorders. The computer-generated numbers were used for the randomization process [19]. We used even and odd numbers to allocate an equal number of participants into both the experimental and the control group, respectively [21]. Thirty participants in each group completed the trial. Group A (experimental group 1) completed resistance sprint training, Group B (experimental group 2) completed plyometric training and Group C (control group) did not complete any training. Pre-and post-testing was carried out at weeks one and six, respectively. The outcome variables were the maximal voluntary isometric contraction ((MVIC) of the quadriceps muscle, the sprint performances (ST), and the Single-Leg Triple Hop (SLTH) test. This was measured between the pre- and post- 6 week training program.

### 2.2. Study Population

A total of 250 University level football players were assessed via a telephone interview. Ninety collegiate football players from two cities (Majmaah and Riyadh) in Saudi Arabia with a mean age of 20.48 years who met the inclusion criteria participated. The minimum sample size was calculated as 92 using a computerized priori t-test (matched pairs) keeping the 5% level of significance and a 95% sampling power for the differences of the outcome scores across two time points (pre- and post-), and assuming a standard deviation of 3.5. Recruitment occurred through a local university, sporting clubs, and the community. The inclusion criteria were young male participants in the age group of between 18 and 25 years. They needed to be participating in any level of football training where frequent sprinting was required. The participants with a current injury or a history of any lower limb surgery that affected the lower limb function, any cardio-respiratory disease, or impairments of the spine or lower extremities were excluded from the study. All the participants included in the study were randomly divided into three groups. The two experimental groups were divided into the resisted sprint training (RST) group (*n* = 30) and the plyometric training (PLT) group (*n* = 30). Participants in the experimental groups undertook training for six weeks on alternate days (i.e., three sessions per week) and the control group (*n* = 30) did not train. No significant between-group baseline or pre- to post-training differences in anthropometrics were observed (see Table 1). Additionally, it is also important to note that none of the participants dropped out in this study. Under the Declaration of Helsinki, participants were informed about the possible risks and benefits of the study, and all participants signed their informed consent before their participation in the study. The Ethical Sub-Committee of the College of Applied Medical Science, Majmaah, Saudi Arabia approved this study (Ethics Number: MUREC-Dec.15/COM-2020/13-2).

### 2.3. Study Interventions

After a familiarization session, both the experimental and control groups underwent a training program for six weeks with three sessions per week. Each of the participants underwent a standardized 10–15 min warm-up, which included 7–8 min of jogging and running, and stretching exercises for 5–6 min. All the participants followed the instructions they had been given for participating in the trial.

#### 2.3.1. Resisted Sprint Training Program (RST)

Sprinting was resisted with a “sled” (a sliding metallic frame) attached with the harness through a rope. Each participant was given resistance with 10% of their body weight to pull during sprinting. Then, the participants performed the sprints [22].
20 m sprints × 3 sets× 3 repetitions20 m sprints × 4 sets × 3 repetitions40 m sprints × 3 sets × 3 repetitions40 m sprints × 4 sets × 3 repetitions50 m sprints × 3 sets × 3 repetitions50 m sprints × 4 sets × 3 repetitions

Participants performed these sprints training in a straight line from the starting to the finishing line with a rest period of 2–3 min in between.

#### 2.3.2. Plyometric Training Program (PT)

Plyometrics completed by the participants included: Bounding: This is the form of plyometric training where enormous strides are used in the running action and extra time is spent in the air. During this, there is an increased hip and knee flexion to cover more distance while the arms swing in a regular sprinting action. For the initial two weeks, the participant performed bounding for 30 m for two repetitions (sets), and after two weeks, three sets of 30 m of bounding with a rest period of 3–4 min.Hurdling: A total of 8 cones with a height of 40 cm were kept in a straight line, 1 m apart for doing plyometric training as hurdling. The participant was instructed to jump over the consecutive cones (hurdles) with both legs. For the initial two weeks, the participant performed 2 sets of hurdling over eight cones. For the next 4 weeks, 3 sets of hurdling were completed over eight cones. The rest period was 2–3 min between each set.Drop jumping: This involved the participant dropping (not jumping) to the ground from a stepper (of height 40 cm) and immediately jumping forward maximally. For the initial two weeks, the participant performed 2 sets of 8 repetitions of drop jumping. For the next 4 weeks, 3 sets of 8 repetitions of drop jumping with a rest time of 2–3 min in between each set were completed.

### 2.4. Outcome Measures

#### 2.4.1. Knee Extensors (Maximal Voluntary Isometric Contraction) Strength (STN) Test

The isometric strength of the quadriceps femoris was measured using a reliable and valid ISOMOVE dynamometer (ISO-MANSW-IT, Tecnobody, Dalmine (BG) Italy). The ISOMOVE system, software version 0.0.1 (ISO-MANSW-IT, Tecnobody) was used to record all isometric data. It provided an accurate assessment of the maximum peak torque of the quadriceps muscles. All the subjects were familiarized with the equipment before the commencement of data collection. Participants were in the sitting position and were stabilized with safety belts across the chest, thighs, and hips to avoid displacements, and the shin pad was adjusted at 5.1 cm (2 inch) superior to the medial malleolus (Figure 2). All the measurements were recorded from the participant’s dominant leg, with the hip and knee angles set at 90 degrees of flexion, as this position results in the most significant torque output [23]. Verbal instruction was given to each participant to keep his/her arm crossed over his/her chest and the participant was given verbal encouragement to motivate them to attain maximum effort during the 5 s contractions. Each test included three consecutive 5 s trials with a 2 min rest between the trials. The mean score of three readings was used for statistical analysis.

#### 2.4.2. Sprint Test (ST)

The sprint test is a reliable and valid way to measure the speed performance. Participants started the test in a standing position with their forward leg placed immediately behind the starting line. Then, on command, they started sprinting with a maximal speed over a 50 m distance. All the performance times (in seconds) were recorded by a handheld stopwatch (XINJIE, SW8-2008) [22], when the foot of the participant touched the finishing line. Next, two sprint test trials with a 5 min recovery period were performed. The best of the two scores (i.e., the lowest timing for a 50 m sprint) were considered to be the pre-test (baseline) scores.

#### 2.4.3. Single-Leg Triple Hop Test (SLTH)

The SLTH test scores from the participant’s performance were measured as the distance covered in three hops, using a measuring tape. The participants started the test by standing on the dominant limb with the toes just behind the starting line, and then the participant completed the three consecutive hops on the same limb. The single-leg triple hop test performance measured the distance covered from the starting point to where the back of the participant’s heel hit the ground (please refer to Figure 3) [24,25]. They performed three trials with a 3 min recovery period. The best of the three scores, (i.e., the maximum distance covered) was taken as the pre-test (baseline) score.

### 2.5. Statistical Analysis

The data analyses were carried out using Statistical Programming for Social Studies SPSS software (SPSS Inc., v.22, IBM, Chicago, IL, USA). A mixed design (3 × 2) two-way analysis of variance (ANOVA) was used to identify the main effect of the treatment on the dependent variables across the 2 time points (pre- and post-), between the groups (RST group, PT group, and control group), and the interaction between group and time. Further, a post hoc analysis (Bonferroni’s multiple comparison test) was used to compare the treatment effect on the dependent variables within each group across the time points and between the groups at 6 weeks post-intervention. In addition, Cohen’s *d* test was used to identify the treatment effect on the dependent variables within each of the groups across the two points and between the groups at 6 weeks post-intervention. Pearson’s correlation test was used to corelate among all the dependent variables. The level of significance (α) was set at 0.05.

## 3. Results

The mean (CI) for the age, height, and weight of all the participants was 20.48 (C1:20.11–20.84 years), 1.72 (1.71–1.73 m) and 64.88 (63.8–65.97 kgs), respectively. The general and baseline clinical characteristics of the participants by treatment group is presented in Table 1. No significant differences were found among the groups with respect to their general and baseline characteristics (*p* > 0.05).

The main effect of the treatment on the dependent variables across the two time points (pre- and post-), between the groups, and the interaction between time and group together with the effect size (η) is presented in Table 2. There was a significant difference across the two time points for the scores of the outcomes STN: F (1,87) = 256.60, *p* < 0.05; ST: F (1,87 = 301.40, *p* < 0.05; and SLTHT: F (1,87) = 267.57. Similarly, the significant differences between the groups averaged across time for the scores of the outcomes was found as STN: F (2,87) = 5.42, *p* < 0.05 and ST: F (2,87) = 4.05, *p* < 0.05. However, an insignificant difference was observed between the groups for the scores of the outcomes SLTHT: F (2,87) = 1.17, *p* > 0.05. There was also a significant interaction between time and group for the scores of the outcomes STN: F (2,87) = 3.69, *p* < 0.05; ST: F (2,87) = 38.18, *p* < 0.05; and SLTHT: F (2,87) = 36.28, *p* < 0.05. 

Pairwise comparisons (using the Bonferroni multiple comparisons test) for the scores of the outcomes within each of the groups across the two time points and between the groups at 6 weeks post-intervention are presented in Table 3 and Table 4, respectively. The results within each group showed that there were significant differences (*p* < 0.05) in the participant’s outcome parameters (STN, ST, and SLTHT) across the two time points of the study.

The between group analysis demonstrated a significant difference (*p* < 0.05) between the control group and the plyometric training group in both the strength and sprint tests (Table 4). However, the sprint test and the single-leg triple hop test demonstrated a significant difference (*p* < 0.05) between the control and resisted sprint training groups (Table 4).

In addition, Pearson’s correlation test revealed that the sprint test and the single-leg triple hop test was positively correlated in all the three groups at 6 weeks post-intervention (Table 5).

## 4. Discussion

The aim of the study was to compare the short-term (i.e., 6 weeks) effects of three different training sessions on the strength test, the sprint test and the single-leg triple hop test for collegiate male football players. A sub-group analysis demonstrated an increase in the strength of the plyometric training group and the single leg triple hop test in the resisted sprint training group. In contrast, the sprint test was improved in both the experimental groups. In post-training, the sprint test was positively correlated with the single leg triple hop performance in all three groups.

In agreement with our findings, previous studies [25,26,27,28] have showed that plyometric training effectively improves both the strength and the performance of young adult male players. According to Arazi [29] and Markovic and Mikulic [30], improvements after plyometric training could be due to neuromuscular adaptations, such as the increased neural drive to the agonist muscles, improvements in intermuscular coordination and changes in muscle size. In addition, hypertrophy of the type II muscle fibers as well as the growth spurt-related increases in muscle coordination and motor unit activation greatly influence the power performance [30,31,32,33]. These adaptations could increase the ability of the muscles to produce greater force during sprinting or jumping. Although the muscle strength of the quadriceps was measured in this study, the changes in the muscle bulk were not measured.

Additionally, both the plyometric and the resisted sprint training groups showed improvements in sprint performance. Similar to strength, the improvement in the sprint performance could be due to enhanced motor neuron excitability together with neuromuscular adaptations. According to Mirzaei [34] this is achieved by utilizing a stretch-shortening cycle and an increased neural drive to the agonist muscles and remarkable plasticity. Our findings are in line with previous studies [26,34,35,36]. Ramirez-Campillo [35] reported that 6 weeks of plyometric training improved the performance in 20 m sprints (Effect size (ES) = −0.5) in 14 year old soccer players. Alternatively, in another study by Ramirez-Campillo [36], it was found that 7 weeks of plyometric training had trivial effects (ES = −0.35) on the 20 m sprint performance of 13 year old soccer players. The reasons for this discrepancy could be due to a change in the rate of training effects between these findings [26,35,37] and the discrepancies reported in the previous studies of youth soccer players might be due to the different individual adaptive responses to plyometric training [38,39].

A single-leg hop is a combination of multi-joint actions, rapid eccentric contractions, and high-velocity concentric muscular contractions that are influenced by the stretch reflex. Although there was a marginal increase in the control and plyometric training group, the 6 weeks resisted sprint training group showed a significant improvement in the single-leg hop test. There are numerous components that influence the performance. It is possible that muscle size enhancements, an increase in the cross-sectional area of the muscle and transitions in the fast-twitch muscle fibers could result in greater hoping or jumping ability gains in young adults. In addition, enhanced neural and motor development and better movement quality and coordination could be another reason for more significant gains [40]. 

In general, jumping ability was significantly improved after PT interventions [41,42]. Previous studies have observed improvements in jumping performance after interventions were applied with PT [43,44,45] or without PT. However, none of the aforementioned studies compared the effects of PT with RST. The current study expands on previous knowledge, showing that the single-leg hop test improvements were more significant in the RST than in the PT group. Additionally, there was a strong correlation between sprint test and SLTH.

In summary, our findings suggest that during a short-term training period, PT achieved an advantage over RST training in improving the strength of the quadriceps. However, both RST and PT training are equally capable of enhancing the neuromechanical capacity (sprint performance only) of collegiate football players. Moreover, all the training modes (i.e., PT and RST) can be similarly and effectively used to enhance the maximum single-leg hop in young collegiate football players. It is important to note that there were no adverse events or side effects of the training reported by the participants.

The current research poses some limitations. First, this study focused on collegiate male footballers; the findings can only be interpreted and applied by those who fall into this category. Second, the results may not benefit the sprinting ability of those who participate in other sports or are of a different gender. Finally, Markovic and Mikulic [30] demonstrated that plyometric training for any age, physical activity, and athletic ability provides the same results. In contrast, Moran [46] found that plyometric training affects young male athletes compared to older adults. Another limitation is that it was assumed that those in the control experimental groups were all of the same ability at baseline testing. However, this cannot be standardized due to the positional differences and the technical abilities of the players [1]. This was taken into account and measures were put into place within the analysis to account for this. As a result, further research needs to be undertaken to look at the impact of these training methods on different athletic abilities, from grassroots to elite athletes.

### Application to Clinical Practice

The current study suggests that resisted sprint and plyometric training could be a potential tool in improving the muscle strength and function in collegiate football players. Additionally, sports-specific plyometrics training will be beneficial for coaches and athletes who are willing to improve both sprint and lower limb functional performance. 

Resisted sprint training with a resistance of 10% of the body weight helped to recruit the hip and knee extensor muscles which resulted in the more significant application of power in a horizontal direction as a sprint, jump, or hop (which are key determinants of performance in any sports) [22].

## 5. Conclusions

In summary, this study revealed that the PT group showed an advantage over the RST group in improving the strength of knee extensors during a short-term training period. Moreover, a combination of either RST or PT were equally capable of enhancing the neuro-mechanical capacities (sprint test only) of collegiate football players; however, they failed to enhance the single-leg triple hop test compared to the control group.

## Figures and Tables

**Figure 1 ijerph-18-06702-f001:**
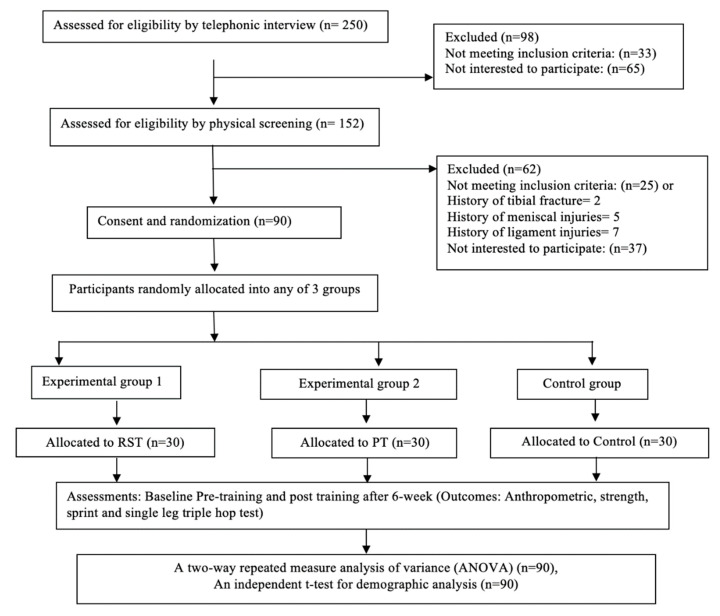
Consolidated Standards of Reporting Trials (CONSORT) diagram showing the flow of participants through each stage of a randomized trial. RST: resisted sprint Training, PT: plyometric Training.

**Figure 2 ijerph-18-06702-f002:**
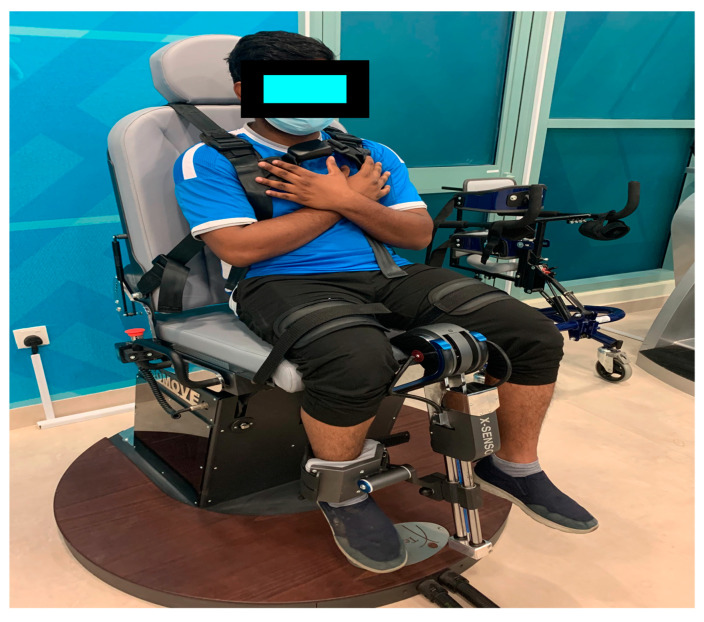
Measurement of the maximal voluntary isometric contraction at 90° using an ISO-MOVE isokinetic dynamometer.

**Figure 3 ijerph-18-06702-f003:**
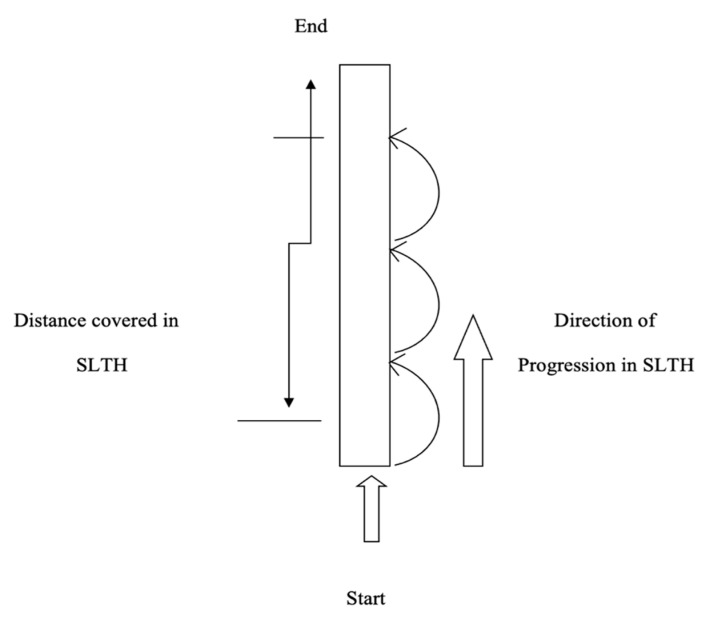
Illustration of Single-Leg Triple Hop test (SLTH).

**Table 1 ijerph-18-06702-t001:** Characteristics of the participants in both the experimental and control groups.

	RST (*n* = 30)	PT (*n* = 30)	Control (*n* = 30)	*p*-Value
Age (years)	20.39 ± 1.77	20.66 ± 1.84	20.39 ± 1.60	0.62
Height (m)	1.73 ± 0.043	1.73 ± 0.050	1.69 ± 0.035	0.62
Body mass (Kg)	63.73 ± 5.2	64.70 ± 4.8	66.23 ± 5.3	0.86
BMI (Kg/m^2^)	21.12 ± 1.74	21.62 ± 1.67	23.07 ± 1.43	0.64

Values are mean values ± standard deviations; BMI–body mass index; RST–Resisted sprint training; PT–Plyometric training; *p* significant at <0.05.

**Table 2 ijerph-18-06702-t002:** The main effect of treatment on the outcomes, within-subject factors across the time (pre- and post-), between-subject factors between the groups (RST vs. PT vs. Control), and the interaction between groups (3) and time (2) using a mixed design 3 × 2 ANOVA test.

Variables	Outcomes	df1	df2	F-Value	*p*-Value	η^2^
Time (2)	STN	1	87	256.599	0.001 *	0.747
	ST	1	87	301.401	0.001 *	0.776
	SLTHT	1	87	267.568	0.001 *	0.755
Time × Groups (2 × 3)	STN	2	87	3.688	0.029 *	0.078
	ST	2	87	38.179	0.001 *	0.467
	SLTHT	2	87	36.276	0.001 *	0.455
Groups (3)	STN	2	87	5.422	0.006 *	0.111
	ST	2	87	4.048	0.02 *	0.085
	SLTHT	2	87	1.177	0.313	0.026

*—Significant value if *p* < 0.05; df: Degree of freedom; η^2^: Eta Squared where η^2^ = 0.01 indicates a small effect; η^2^ = 0.06 indicates a medium effect; η^2^ = 0.14 indicates a large effect.

**Table 3 ijerph-18-06702-t003:** Pairwise comparisons for the scores of the outcomes of the strength (STN), resisted sprint (ST), and muscle performances (SLTHT) across two time points within each group using Bonferroni’s multiple comparison test. Cohen’s *d* test was applied for measuring the effect size between the two time points.

Outcomes	Groups	Pre intervention	Post Intervention	Time (Pre-Post)	*p*-Value	Cohen’s *d*
STN (∆MD ± SE)	RST	13.73 ± 0.77	14.98 ± 0.67	−1.248 ± 0.155	0.001 *	1.63 ˆ
	PT	13.47 ± 0.94	51.26 ± 0.91	−1.782 ± 0.155	0.001 *	2.38 ˆ
	Control	13.22 ± 0.66	14.51 ± 0.78	−1.283 ± 0.155	0.001 *	2.26 ˆ
ST (∆MD ± SE)	RST	9.19 ± 0.58	8.38 ± 0.75	0.813 ± 0.053	0.001 *	1.21 ˆ
	PT	9.22 ± 0.40	8.61 ± 0.50	0.618 ± 0.053	0.001 *	1.36 ˆ
	Control	9.24 ± 0.45	9.07 ± 0.45	0.171 ± 0.053	0.002 *	0.38
SLTHT (∆MD ± SE)	RST	501.30 ± 54.50	548.03 ± 49.90	46.73 ± 3.118	0.001 *	0.89
	PT	500.00 ± 50.74	532.13 ± 51.860	32.133 ± 3.118	0.001 *	0.61
	Control	499.90 ± 51.13	509.37 ± 50.41	−9.467 ± 3.118	0.001 *	0.18

*—Significant value if *p* < 0.05; ˆ—Large effect size if Cohen’s *d* value > 0.8; I: Baseline score; J: Post-intervention scores at 6 weeks; ∆MD: Mean differences); SE: Standard error; RST: Resisted sprint training; PT: Plyometric training; ∆MD: Mean differences; STN: Strength; ST: Resisted sprint SLTHT: Single leg triple hop test.

**Table 4 ijerph-18-06702-t004:** Pairwise comparisons of the post test scores (at 6 weeks) for the outcomes of strength (STN), resisted sprint (ST), and muscle performance (SLTHT) between the groups using Bonferroni’s post hoc test. Cohen’s *d* test was applied for measuring the effect size between the two groups.

Outcomes	Treatment Groups		∆MD ± SE	*p*-Value	Cohen’s d
STN	RST	PT	−0.276 ± 0.205	0.554	0.439
		Control	0.473 ± 0.205	0.070	0.760
	PT	Control	0.750 ± 0.205	0.001 *	1.577 ˆ
ST	RST	PT	−0.235 ± 0.150	0.366	0.361
		Control	−0.695 ± 0.150	0.001 *	1.132 ˆ
	PT	Control	−0.460 ± 0.150	0.009 *	0.989 ˆ
SLTHT	RST	PT	15.90 ± 13.10	1.000	0.312
		Control	38.67 ± 13.10	0.012 *	0.771
	PT	Control	22.77 ± 13.10	0.269	0.446

*—Significant value if *p* < 0.05; ˆ—Large effect size if Cohen’s d value > 0.8; RST: Resisted sprint training; PT: Plyometric training; ∆MD: Mean differences; SE: Standard error; STN: Strength; ST: Resisted sprint; SLTHT: Single leg triple hop test.

**Table 5 ijerph-18-06702-t005:** Correlation of strength, sprint test, and single leg triple hop test at post-intervention.

	STN Po	ST Po	SLTH Po
STN Po	1	−0.26 (0.13)	0.039 (0.71)
ST Po	-	1	−0.36 (0.00) *
		Plyometric Training Group	Control Group
Resisted Sprint Training		0.79	0.05 *
Plyometric Training Group		–	0.20

*—*p* ≤ 0.05.

## Data Availability

All data sets related to the results of this study are available from the primary author on request.

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
