# Peer review of "Effect of Resisted Sprint and Plyometric Training on Lower Limb Functional Performance in Collegiate Male Football Players: A Randomised Control Trial"

_ijerph, 2021, doi:10.3390/ijerph18136702_

Round 1

Reviewer 1 Report

Introduction
L42 What types of performance? Can you be more specific.
L43 Yes, these topics have been heavily researched but plenty of research exists within resisted sprinting and plyometrics. Especially in youth populations for the latter.
L44 Unclear on where these findings are from? Who/what papers showed good results? Also, could you include an example of good results? Evidence of pre and post for a certain metric.
L57 Could you provide a more relevant example fitting with your current population, finding research in football which uses plyometric or resisted sprint would be more fitting and still sets up the question which is more effective?
L63 Could you find an injury prevention study for the study population instead?
L73 Research question is directed towards adult football players, yet the study cohort is collegiate players.

Methods
L89 Reword Pre-and post-testing was done at first and sixth weeks respectively, ‘Pre and post testing was carried out at weeks one and six respectively’.
L94 Figure description missing the word “training” following plyometric.
L97 Collegiate level sample, earlier in “research question” and “objectives” sections professional level is mentioned. Although may provide inferences, this study would not directly translate to the professional level.
L103-135 Could include the week next to each of the prescriptions, just for some clarity.
L169 Validity and reliability of this measurement? May want to acknowledge somewhere that the method for measuring sprint times may not be the most accurate method.

L188 should this not be a two-way anova (time by group)? Also, you need to detail the thresholds for the Cohens. You might also add in a between group effect size

Results
The data in tables 2-5 is not clear at all.

L215You need to justify why you set a large effect size at 0.6, this is the threshold for moderate.

Discussion
L226 As not technically under the banner of “youth” I don’t think you need to say they are young adult males, can just refer to them as collegiate.
L236 Not sure growth spurt related changes fit in here, I would stick to the original points of what adaptation plyometric training provides.
L240 Don’t think it’s needed as all you are trying to demonstrate is force changes, can remove that sentence in my opinion.
L255 Could provide more information about why this test was chosen, does a lower score correlate with an increased injury risk? What are you hoping to see from this test?
L258-260 Can you find any evidence to infer this?
L272 As you didn’t measure the effect of the combination of the two methods in the same session, I think you should remove this. From this data we do not know what effect would be observed.
L273 CT? meaning the control? Seems like it’s suggestion that either of the two interventions or doing nothing can similarly improve Strength, sprint and single leg hop?
L275-276 I don’t know what you are getting at here? I haven’t seen changes in anthropometric measures reported?
L277-280 You can remove that limitation in my opinion. Yes, it doesn’t give direct applications for other populations, but it’s not meant to, it’s there to give application to the cohort studied.
L280-283 Not sure what this supporting evidence is there to achieve? Again, the study is not about other cohorts it is about the effect on the collegiate sample.
L289-293 Again can remove this limitation in my opinion, the point for further research is valid saying this study was short-term what about long term. However, the study being short term is not a limitation as it answers the question of what adaptations can be seen over 6 weeks.
L298 Collegiate athletes instead of young, this may confuse a reader
L301-304 Evidence for the increased activation of the knee and hip extensors resulting in more horizontal force? 

Author Response

The authors would like to express sincere thanks to the reviewers for their time and effort in reviewing our manuscript. We are thankful to the reviewers for their constructive and valuable comments/suggestions to improve the quality of our manuscript. Each individual comment/suggestion has been addressed as appropriate, and the revised manuscript has been submitted for your consideration. All changes in line with the reviewers’ comments/suggestions have been highlighted with track changes in the revised manuscript. The point-to-point responses to all comments/suggestions are listed in the attached file.

Reviewer 2 Report

ABSTRACT

Line 23: add Description of the trial design (such as parallel, cluster, non-inferiority)

Line 23: Eligibility criteria for participants and the settings where the data were collected

Line 27: inform whether participants, caregivers, and those assessing the outcomes were blinded to group assignment

Lines 28-31: For the primary outcome, a result for each group and the estimated effect size and its precision

Line 31: Add important adverse events or side effects

Line 33: Add the registration number and name of the trial register

INTRODUCTION

Lines 38-44: this paragraph is hard to follow in the rationale. My suggestion is to start with the importance of reactive strength, strength, or sprinting for overall performance in soccer. Then, move for the importance of applying plyometric training and resisted sprinting and why.

Lines 42-44: generally focused? May systematic reviews and meta-analysis have been conducted in soccer and some original research in the resisted sprint.

Line 44-45: which findings? Which articles? Why?

Lines 45-47: why talk about running, while many articles about these topics were published in soccer?

Lines 48-56: mechanisms and neuromuscular changes occurring in plyometric training should be described to justify the changes and the expectations.

Lines 57-71: again, the rationale is not easy to follow.

Overall recommendation: the structure of the introduction should be extensively improved. Start talking about the importance of strength and sprinting for soccer players. Explain how these qualities may support performance and make the difference between players. After that, introduce plyometric training and the mechanisms that explain the advantage of soccer. A similar approach for the resisted sprint. After these two introductions, talk about combined training, namely explaining why and how to combine slide sprinting and plyometric training. Finally, add a paragraph of the statement of contribution and presents the objectives.

METHODS

Line 86 Description of trial design (such as parallel, factorial) including allocation ratio. Add Important changes to methods after trial commencement (such as eligibility criteria), with reasons

Line 87: add the method used to generate the random allocation sequence. Add the type of randomization; details of any restriction (such as blocking and block size). Add mechanism used to implement the random allocation sequence (such as sequentially numbered containers), describing any steps taken to conceal the sequence until interventions were assigned. Inform about who generated the random allocation sequence, who enrolled participants, and who assigned participants to interventions

Line 89: If done, who was blinded after assignment to interventions (for example, participants, care providers, those assessing outcomes) and how

Line 98-100: How sample size was determined?

Line 100: Eligibility criteria for participants (clearly define). Also, add settings and locations where the data were collected

Line 122: add information about the adherence of participants in both groups across the intervention period.

Line 122: add information about the number of training (field-based) and matches during the period. Add information about the rest between sessions.

Line 138: add information about the dose, of plyometric training, namely number of repetitions, intensity, type of surface, the overall number of jumps, density (work to rest ratio), sequence of exercises, and variation across the weeks.

Line 156: add information about the control group and activities made.

Line 157: add information about the field-based training made by all the players, and how this may create a bias for the study

Line 156: contextualize the data collection, namely period before (rest), sequence of tests in the same day, the protocol of warm-up, the period of the day, temperature and relative humidity, type of pitch, familiarization, reliability of the observers, and instruments used.

Line 173: stopwatch? How was the reliability ensured? Critical question.

Line 182: how the reliability of the observer was tested?

Line 187: add information about normality and homogeneity of the sample

IMPORTANT: The statistical approach should be re-calculated. The design implies a mixed ANOVA - A mixed ANOVA compares the mean differences between groups that have been split on two "factors" (also known as independent variables), where one factor is a "within-subjects" factor (time/ post-pre) and the other factor is a "between-subjects" factor (groups).

RESULTS

Should be re-written - The statistical approach should be re-calculated. The design implies a mixed ANOVA - A mixed ANOVA compares the mean differences between groups that have been split on two "factors" (also known as independent variables), where one factor is a "within-subjects" factor (time/ post-pre) and the other factor is a "between-subjects" factor (groups).

Additionally, add a figure of pos-pre intervention analysis for each of the outcomes strictly following the example presented in this article “Nimphius, S., & Jordan, M. J. (2020). Show Me the Data, Jerry! Data Visualization and Transparency. International Journal of Sports Physiology and Performance15(10), 1353-1355.” – figure 2 (presenting variation for each of the participants and for the mean of the population)

Also add: All important harms or unintended effects in each group

DISCUSSION

Generalisability (external validity, applicability) of the trial findings is missing.

At the end of the discussion add: Registration number and name of the trial registry and Where the full trial protocol can be accessed, if available

Author Response

The authors would like to express sincere thanks to the reviewers for their time and effort in reviewing our manuscript. We are thankful to the reviewers for their constructive and valuable comments/suggestions to improve the quality of our manuscript. Each individual comment/suggestion has been addressed as appropriate, and the revised manuscript has been submitted for your consideration. All changes in line with the reviewers’ comments/suggestions have been highlighted with track changes in the revised manuscript. The point-to-point responses to all comments/suggestions are in the attached file.

Reviewer 3 Report

Title: Effect of Resisted Sprint and Plyometric Training on Lower Limb Functional Performance in Young Adult Male Football 4 Players: A Randomised Control Trial

Major concerns

- while this is a reasonably well thought-out and conducted study, the final information derived from this study is nothing new and no novel data in many aspects such as population type, sample size, type of training interventions, etc,. There have been many, many and many studies that have shown the same outcomes – so that is little value-add info that is being provided by this study. There is clearly a need to better understand the mechanisms underlying these training-induced adaptations rather merely provide more and more evidence that both PT and RST of training works in enhancing exercise performance.

- the lack of information on the types of training that the players gone through (other than the RST and PT interventions that investigators are imposing on the players) such as what football training or any other types of S&C or physical training that the groups have had undergone during the 6 weeks of intervention. The reason for this query is simple – how do the authors know that the others types of training did not cause the differences observed in the present study outcomes. Clearly it is important to make sure that other types of training were equivalent between the 3 groups. For example, in Line 110-111, it was mentioned that the control group “was instructed to maintain regular activities and avoid any strenuous physical activity during the study” -  how can the control group avoid strenuous exercises when they are collegiate players in training. Don’t the players maintain their normal training which is surely physically strenuous on some occasion throughput the 6 weeks period! The authors need to  provide evidence the other trainings (such as football and other strength and conditioning trainings) were equivalent between the 3 groups.

- the study reported showed that leg strength was enhanced to a greater extent in PT relative to RST and single leg hop test (power) was enhanced to a greater extent in RST relative to PT. How do the authors know that these differences was not to the  differences in accumulated “physically load” between the RST and PT training sessions that the players have encountered during the 6 weeks period. In short, could the differences in training-induced adaptations observed be due to the differences in the volume/intensity rather differences in the 2 types of training that the 2 groups have undergone. It would have been ideal if some measure of training load, e.g., session RPE have been taken at the end of each session during the RST and PT training to see if the 2 types of training were of equivalent physical load.

- the training interventions were not well described and insufficient details were provided. For example, the RST line 130-135, it was not clear whether these sprints were done during a single session or each line is representing a week of training; and what does “x 3 x 3” mean? Is it 3 repeats x 3 sets?

- Another concern is the use of stopwatches to manually timed the sprints. In this modern-age, I think the used of stopwatches to assess accurately sprinting times should be eliminated altogether – because the human error made is larger than the expected changes made due from training intervention!

- Statistically analyses. A 2-way ANOVA repeated measure (with 3 groups and 2 time points) should be the gold standard of analysis to compare the differences between group in the present study. Also, to use effects size differences between groups to add to the statistics analysis. Also, the coefficient of variation or smallest worthwhile change of all the three tests should be provided here. This will allow the reader to appreciate the “minimum threshold magnitude of the change”.

Minor issues

- please get an English-speaking native speaker to vet through your manuscript. Some sentences were awkward and does not make sense.

- I assume that the authors deemed University collegiate levels as “professional athletes”? Please justify this.

- Line 73. The research question should be written and embedded in the discussion section and not written separately. Note that this manuscript should be written as a thesis format.

- Line 81-83. The hypotheses should be written as a paragraph discussion. The manuscript is not a thesis format.

- Line 99-100. What do you mean by “score differences of 2 points”? there should be a unit of measure here?

- Line 107. Should be “PT” rather than “PLT”.

- Line 129. The sentence sis incomplete.

- Line 144 Is it repetitions or sets? Pleas be clear.

- Line 47. Not “cms” should be “cm”.

- Line 157. Why complicate things by using “STN”?

Author Response

(The authors gave the same response as above.)

Round 2

Reviewer 2 Report

The article was substantially improved. Based on that, it can be accepted.

Author Response

We are thankful for the reviewer for his/her comments. We really appreciate the feedback and query. 

As there is no comment raised, nothing to be responded on.

Reviewer 3 Report

- Authors mentioned that all 3 groups did not do any other forms of training over the 6 weeks because of Covic. My query then, how the RST and PLT training sessions were conducted during this same period? - via virtual? if that's the case, how does the authors ensure that the players performed as exactly as what the authors have had wanted them to do during these sessions?

Author Response

We are thankful for the reviewer for his/her comments. We really appreciate the feedback and query. 

In response to the reviewer query we would like to confirm that the training was undertaken face to face under the supervision of the researcher and not virtually by following all the Covid-19 guidelines. We would like to reiterate that all the participants did not involved in group training or played any matches during our data collection period, excluding our own exercise protocol. Hope this helps. If you need any more information on this please let us know.